# Dissection of the Multichannel Reaction O(^3^P) + C_2_H_2_: Differential Cross-Sections and Product Energy Distributions

**DOI:** 10.3390/molecules27030754

**Published:** 2022-01-24

**Authors:** Shuwen Zhang, Qixin Chen, Junxiang Zuo, Xixi Hu, Daiqian Xie

**Affiliations:** 1Institute of Theoretical and Computational Chemistry, Key Laboratory of Mesoscopic Chemistry, School of Chemistry and Chemical Engineering, Nanjing University, Nanjing 210023, China; dz1924043@smail.nju.edu.cn (S.Z.); mg1724007@smail.nju.edu.cn (Q.C.); 2Department of Chemistry and Chemical Biology, University of New Mexico, Albuquerque, NM 87131, USA; jxzuo@unm.edu; 3Kuang Yaming Honors School, Institute for Brain Sciences, Nanjing University, Nanjing 210023, China

**Keywords:** reaction dynamics, quasi-classical trajectory calculation, differential cross-section, product energy distribution, complex-forming reaction, reaction mechanism

## Abstract

The O(^3^P) + C_2_H_2_ reaction plays an important role in hydrocarbon combustion. It has two primary competing channels: H + HCCO (ketenyl) and CO + CH_2_ (triplet methylene). To further understand the microscopic dynamic mechanism of this reaction, we report here a detailed quasi-classical trajectory study of the O(^3^P) + C_2_H_2_ reaction on the recently developed full-dimensional potential energy surface (PES). The entrance barrier TS1 is the rate-limiting barrier in the reaction. The translation of reactants can greatly promote reactivity, due to strong coupling with the reaction coordinate at TS1. The O(^3^P) + C_2_H_2_ reaction progress through a complex-forming mechanism, in which the intermediate HCCHO lives at least through the duration of a rotational period. The energy redistribution takes place during the creation of the long-lived high vibrationally (and rotationally) excited HCCHO in the reaction. The product energy partitioning of the two channels and CO vibrational distributions agree with experimental data, and the vibrational state distributions of all modes of products present a Boltzmann-like distribution.

## 1. Introduction

Acetylene is an important intermediate species in most hydrocarbon flames [1,2,3]. Acetylene reactions, as part of the oxidation mechanism of the other hydrocarbon fuels, are crucial in the formation of large amounts of ions, other higher hydrocarbons, and soot. Detailed dynamic information on the key elementary reaction involving acetylene is essential to better understand the mechanisms as well as to optimize the combustion process. A dominant pathway for consumption of acetylene is initiated by bonding of the electrophilic O(^3^P) atom to the unsaturated bonds of acetylene, resulting in the following two primary product channels:(R1)$$O{(}^{3}P)+C_{2}H_{2}\left(X^{1}\Sigma_{g}^{+}\right)\to H{(}^{2}S_{1/2})+\mathrm{HCCO}\left(X^{2}A^{\prime \prime }\right)$$
(R2)$$O{(}^{3}P)+C_{2}H_{2}\left(X^{1}\Sigma_{g}^{+}\right)\to \mathrm{CO}\left(X^{1}\Sigma^{+}\right)+{\mathrm{CH}}_{2}\left(X^{3}B_{1}\right)$$

Due to its vital significance in combustion chemistry, this reaction has attracted great attention. The kinetic and dynamic properties for this reaction have been measured and calculated over a wide temperature range using various techniques [4,5,6,7,8,9,10,11,12,13,14,15,16,17,18,19,20,21,22,23,24,25,26,27,28]. Until recently, the experimental rate constants and branching ratios (BRs) have been confirmed by quasi-classical trajectory calculations based on the global potential energy surface (PES) at UCCSD(T)-F12b/VTZ-F12 level developed by our group [29]. The reaction pathways are quite complex, consisting of seven transition states and four intermediates, as shown in Figure 1. However, there are fewer experimental and theoretical studies focusing on other dynamic information, such as differential cross-sections and product energy distributions, which are also very important to gain insight into such a multiwell multichannel reaction.

The crossed molecular beam (CMB) experiments of the O(^3^P) + C_2_H_2_ reaction have been investigated by several groups [10,14,19,21,22]. Clemo et al. [10] determined the translational-energy dependence of the total cross-section by using a supersonic beam of O atoms seeded in He. Schmoltner et al. [14] found that the reaction proceeds through a long-lived intermediate. In 2014, Leonori et al. [22] revealed that the reaction proceeds through a complex-forming mechanism. According to our previous QCT analysis, the complex-forming mechanism is the key to understanding the dynamics of the title reaction. The reaction process is likely to undergo energy redistribution. In addition, the vibrational and rotational distributions of product CO in channel R2 were measured via different spectroscopy experiments. Three groups [20,30,31] had observed the vibrational excited CO product from the channel R2 and concluded that the vibrational states of CO present an approximate Boltzmann distribution, which can be quantitatively accounted for by a simple statistical model [31,32]. However, Nguyen et al. [25] had found that product yields calculated from conventional RRKM theory depart from the experimental branching fractions (~80% R1 and ~20% R2) [22] and indicated that the rate of internal-rotation isomerization (Int2↔Int3) was hampered by non-statistical energy partitioning, which implied that the statistical RRKM theory failed for certain intermediate steps. In our previous work [29], we classified trajectories by the propagation time and found that the two complexes (Int2 and Int3) could achieve a microcanonical equilibrium in slow reactions, but such an equilibrium was not established in fast reactions. In light of these studies, it is clear that the available energy is redistributed at all internal degrees of freedom. But in O(^3^P) + C_2_H_2_ reactions, the relation between energy distribution and dynamic process is not yet clear. More information about differential cross-sections and product energy distributions is needed to fully understand the complex-forming mechanism of this reaction.

To improve our understanding of the kinetics of the title reaction, in this work we reported a detailed quasi-classical trajectory (QCT) study of the O(^3^P) + C_2_H_2_ reaction on the recently developed PIP-NN PES [29], focusing on the complex-forming mechanism and energy distribution in the reaction. We investigated the effect on reactivity by vibrational excitation of C_2_H_2_ and extracted the final state information of the products for analyzing the effects of dynamics and statistics on the reaction. The paper is organized as follows: The computational details in QCT calculations are presented in Section 2. The results and discussion are in Section 3. Finally, conclusions are given in Section 4.

## 2. Methods

In this work, all QCT calculations were carried out using the VENUS program package [33,34] to investigate the mechanisms at the collision energies ranging from 5 kcal/mol to 13.05 kcal/mol. After testing with a small set of trajectories, the maximum impact parameter ($b_{max}$) was set to 2.5 Å. The impact parameter *b* was selected randomly from the distribution $b_{max}\sqrt{r}$, where *r* is a random number uniformly distributed from 0 to 1. Roughly 400,000 trajectories were run based on the PIP-NN PES with a time step of 0.1 fs and a maximum time of 10.0 ps, to converge the total energy within 0.04 kcal/mol in the propagation. The trajectories were initiated at O-C distances of 10 Å and stopped when the products (H + HCCO or CO + CH_2_) reached a separation of 8 Å or reactants were separated by 10.5 Å. A few trajectories (~1%) that failed to converge energy or reaction time longer than 10.0 ps were discarded. In the two channels, most of the reactive trajectories obeyed the quantum mechanical zero-point energy (ZPE) criteria. To avoid rising to unphysical results by forcing individual trajectories to satisfy the quantum mechanical ZPE condition, the “passive” method [35] was applied to both product channels, in which the trajectories were discarded for the total vibrational energy less than the ZPE.

The integral cross-section (ICS) was calculated according to $\sigma_{r}={\pi b}_{\max}{}^{2}P_{r}$, where the reaction probability $P_{r}$ is defined as the ratio between the numbers of the reactive (*N*_r_) and total (*N*_total_) trajectories at a specified initial condition. The statistical error is given by
(1)$$\Delta =\sqrt{\left(N_{\mathrm{total}}-N_{r}\right)/N_{\mathrm{total}}N_{r}\;}$$
which was smaller than 0.07 in this work.

The differential cross-section (DCS) was obtained by
(2)$$\frac{{d\sigma }_{r}}{d\Omega }=\frac{\sigma_{r}P_{r}\left(\theta \right)}{2\pi \;\sin\left(\theta \right)}$$

The scattering angle *θ* is defined as the angle between the velocity vectors ${\vec{v}}_{i}={\vec{v}}_{O}-{\vec{v}}_{C_{2}H_{2}}$ and ${\vec{v}}_{f}={\vec{v}}_{\mathrm{HCCO}}-{\vec{v}}_{H}$ (for R1 channel) or ${\vec{v}}_{f}={\vec{v}}_{{\mathrm{CH}}_{2}}-{\vec{v}}_{\mathrm{CO}}$ (for R2 channel):(3)$$\theta =\arccos\left(\frac{{\vec{v}}_{i}\cdot {\vec{v}}_{f}}{\left|{\vec{v}}_{i}\right|\left|{\vec{v}}_{f}\right|}\right)$$

Notably, the angle of 0° corresponds to forward scattering, while 180° corresponds to backward scattering.

Vibrational quantum numbers of diatomic product molecules can be calculated by Einstein–Brillouin–Keller (EBK) semiclassical quantization of the action integral, which is capable of handling anharmonicity for diatom molecules [36].

For polyatomic products, the normal mode analysis (NMA) method [37,38,39,40,41] based on the harmonic approximation and the decoupling of all degrees of freedom was applied to gain quantum-like vibrational states. In the NMA method, the coordinates and momenta, extracted from one step of the last period that has the minimum potential energy of each reactive trajectory, are taken as input. The kinetic and potential energies of each normal mode are computed by projecting the displacement and momentum matrices onto the respective normal mode space.

The vibrational quantum numbers were determined from the non-integer action variables by binning. Two binning methods, namely histogram binning (HB) and Gaussian binning (GB) [42,43,44], were generally implemented. In the HB method, all trajectories are considered with weight unit and action variable rounded to the nearest integer value, and the probability of state *n* is
(4)$$\;P_{\mathrm{HB}}\left(n\right)=\frac{N\left(n\right)}{N_{\mathrm{traj}}}$$
where N(n) is the number of the products in a particular vibrational state *n* of the reactive trajectories and $N_{\mathrm{traj}}$ is the total number of the reactive trajectories.

For the GB method, the energy-based GB (1GB) method proposed by Czakó and Bowman was employed in this work [37,41]. The Gaussian weight factor is calculated for each normal mode of the *p*th product in a given vibrational state *n*, as
(5)$$\;G_{p}\left(n\right)=\frac{\beta }{\sqrt{\pi }}\exp\left[-\beta^{2}{\left(\frac{\left[E\left(n_{p}^{\prime }\right)-E\left(n\right)\right]}{\left[2E\left(0\right)\right]}\right)}^{2}\right],\;p=1,\;2,\cdots ,\;N\left(n\right),$$
where $n^{\prime }$ is the non-integer classical action variable, and $\beta =2\sqrt{\ln2}/\delta$ is a positive, real parameter. δ is the full width at half-maximum that was taken as 0.2. In this way, the probability of the vibrational state n is given by
(6)$$P_{\mathrm{GB}}\left(n\right)=\frac{\sum_{p=1}^{N\left(n\right)}G_{p}\left(n\right)}{N_{\mathrm{traj}}}$$

The study of Czakó and Bowman showed that 1GB could be especially useful at the regions near the energetic threshold of certain product states, while HB could not reproduce this energetic threshold [37]. The 1GB method could obtain a more realistic energetic description of the finial analysis than the HB method. In this work, population of the vibrational state of the polyatomic products were calculated by the 1GB method, and the population of the vibrational state of the diatomic product was calculated by the HB method.

## 3. Results and Discussion

Figure 1 displays geometries of all stationary points and their energies relative to the reactant asymptote on the PIP-NN PES in our previous work [29]. The addition of the oxygen atom onto a C atom in acetylene takes place on the ground state PES via a shallow pre-reaction well (Int1) and a barrier (TS1), leading to intermediate complex Int2 (or Int3). TS1 is the bottleneck in the reaction, which has a great effect on reactivity. Because the internal rotation barrier is small, a significant amount of available energy will flow into other internal degrees of freedom of the complex via rapid Int2↔Int3 isomerization. Once the energy redistribution occurs, the energy along the reaction coordinate is insufficient for the molecule to overcome the following barrier. Therefore, a microcanonical equilibrium between the two isomers is established [25,29,45]. After that, confronted with different kinds of exit channel barriers, the complex decomposes into different fragments.

### 3.1. Mode Specificity on Reactivity

Focusing on the mode specificity for direct reactions, many studies showed the differing capacity of reactant modes in overcoming barriers. Nevertheless, few studies discussed how the vibration excitation influences the reactivity of complex-forming reactions. In this part, we will discuss this topic in detail.

The reactant C_2_H_2_ is a linear molecule, which has seven vibrational normal modes. Table 1 lists the acetylene harmonic vibrational frequencies obtained using PIP-NN PES. The integral cross-sections (ICSs) at collision energies, ranging from 5 kcal/mol to 13.05 kcal/mol together with vibrational excitation of C_2_H_2_ at *E*_c_ = 8.22 kcal/mol, are presented in Figure 2. Every vibrational mode excitation makes little difference on reactivity, while translation has a great effect on reactivity. In addition, the value of BR is hardly influenced by specific mode excitations, and R1 channel is always the dominant reaction pathway accounting for 80–90% (see Figure 3).

To gain further insight into the effect of reactant modes on the reaction, we used the sudden vector projection (SVP) model to analyze TS1 [46,47]. The SVP model assumes the collision time is very short, and the relative efficacy of a particular motion can be estimated by the projection of the corresponding normal mode vector onto the vector representing the reaction coordinate in the sudden limit. The calculated SVP values for the reactant at TS1 are listed in Table 1. It is clear that the projection value (0.723) of translation is large, which indicates that translation has strong coupling with the reaction coordinate, resulting in a significant influence on the reactivity. Except for C-H symmetrical and asymmetrical bending modes, every C_2_H_2_ vibrational mode has essentially no projection onto the reaction coordinate, which means these modes have little effect on reactivity. The SVP values of C-H bending modes (*ν*_4_, *ν*_5_, *ν*_6_, *ν*_7_) are not small, but excitation of C-H bending modes shows a weak relationship with the ICSs. We also calculated the ICSs for the O + C_2_H_2_ reaction at the lower translational energy (*E*_c_ = 7.00 kcal/mol). The values of ICS are 0.653 Å^2^ (C_2_H_2_ (*ν* = 0)) and 0.744 Å^2^ (C_2_H_2_ (*ν*_6_/*ν*_7_ = 1)) for the H + HCCO channel and 0.105 Å^2^ (C_2_H_2_ (*ν* = 0)) and 0.116 Å^2^ (C_2_H_2_ (*ν*_6_/*ν*_7_ = 1)) for the CH_2_ + CO channel, respectively. These values indicate that the vibrational enhancement effect is indeed weak in this reaction. We speculated that the steric effect of the H atom hinders the O atom from attacking the C atom, and the steric effect will be enhanced by the excitation of C-H bending modes.

The results indicate that vibration mode excitation not only has little effect on reactivity, but also has nearly no influence on BR, and this result is only possible because of the deep well (Int2 or Int3) in the potential energy along the reaction coordinate. Figure 4 shows two typical reactive trajectories for each channel. After reactants cross the first barrier (TS1), the trajectory enters the region of potential well. Int2 and Int3 expend the lion’s share of the lifetime of the complex. In this period, some fraction of the total energy can flow into the other degrees of freedom, and the process is accompanied by a rapid isomerization between Int2 and Int3 [45]. Thus, energy will redistribute between internal modes of the long-lived complex, which will to some extent erase the information of reactants. Moreover, the total angular momentum is conserved in the reaction. For O(^3^P) + C_2_H_2_, the total angular momentum is approximately equal to the orbital angular momentum of the reactants. The complex will receive some rotational excitation, which has a decisive effect on the shape of the differential cross-section.

### 3.2. Differential Cross-Sections and Product Energy Fractions

The shape of differential cross-sections (DCSs) can be related to the reaction mechanism. Associated with a product energy fraction, we can understand that the energy flows in different modes from a microscopic perspective. The calculated DCSs of R1 and R2 at *E*_c_ = 8.22 kcal/mol are displayed in Figure 5, compared with the previous CMB experiments. In the H + HCCO channel, the shape of DCS is basically isotropic except for slight distinctions, which include sideways bias, backward-forward symmetric bias, and forward bias in experiments of Clemo et al., Schmoltner et al., and Leonori et al., respectively [10,14,22]. Our result exhibits a slight sideways scattering, same as the result of Clemo et al. [10], related to the separate ways. This shape resembles the sideways scattering of the CH_2_CHF product observed in the F(^2^P) + C_2_H_4_ reaction and the CH_2_CHO product observed in the O(^3^P) + C_2_H_4_ reaction [48,49]. In this channel, the fragments separated randomly after many rotational periods, and the distribution of products was uniform in this collision plane. The hydrogen atom was emitted nearly orthogonal to the plane of the heavy C-C-O atoms at the decomposing transition state, so the scattering angle of the R1 channel was almost uniformly distributed in the whole space with a slight sideways scattering (see Figure 5c) [50]. In the CH_2_ + CO channel, the shape of DCS had a backward-forward distribution in the experiment of Schmoltner et al. [14] and the calculation of this work. After many rotational periods, intermediate CH_2_CO departed on the same plane of the C-C-O heavy atoms, and thus the DCS exhibited polarized distribution (see Figure 5d) [50]. The result in the experiment by Leonori et al. [22] was backward-forward distribution with a significant forward bias, which may have been caused by the process singlet-CH_2_CO→CH_2_(^1^A_1_) + CO via intersystem crossing. Rajak and Maiti studied the reaction O(^3^P) + C_2_H_2_ by a direct dynamics surface hopping method, and they observed that the fraction was about 1/3 to 1/2 of forming the singlet products for the CH_2_ + CO channel at the collision energy in the range of 8.2~13.1kcal/mol [26,28]. From the above results, we speculate that the DCS of O(^3^P) + C_2_H_2_→CH_2_(^1^A_1_) + CO is forward scattering, which is the research target in the future.

The calculated product energy partitioning of R1 and R2 is presented in Table 2 and Table 3, compared with experimental results. In the H + HCCO channel, the theoretical predictions agree with the experimental results. A great amount of available energy flows into the product translation and vibration of HCCO. The fraction of HCCO vibrational energy accounts for more than 40%, and it increases for the reaction at vibrational excitation of C_2_H_2_. In the CH_2_ + CO channel, the largest fraction of energy is deposited as translation for ~40%, which agrees with the CMB experiments of Schmoltner et al. [14] and Leonori et al. [22]. The fraction of the total available energy channeled into vibrational energy of CO was investigated by Shaub et al. [31] and Chikan et al. [20] by using the CO laser resonant absorption technique and Fourier transform infrared emission spectroscopy, respectively. Although the collision energy was not reported in their works, their data (6.1 ± 1.3% and 7.4%) are close to the calculated results in this work. These results indicate that the available energy distributed in all degrees of freedom seems to reach equilibrium. In addition, the result of the fraction of CO vibrational energy in experiments of Huang et al. [18] disagrees with the result of Shaub et al. [31] and Chikan et al. [20]. In experiments by Huang et al., a large population of CO stays at vibrational ground state, and only 1% or less of the available energy is channeled into rotational energy of CO [18], which both disagree with our calculations.

Beyond that, the product translational energy distribution (see Figure 6) and the large fraction of translational energy indicate the existence of an exit channel barrier in each channel, which can also be confirmed by the DCS results. The differences of the translational energy distribution in the R2 channel between theory and experiment may be caused by forming the singlet products.

### 3.3. Vibrational State Distributions of Products

To further understand the vibrational energy partition in different modes, the vibrational state distributions of HCCO in the R1 channel and CO, CH_2_ in the R2 channel were calculated. Because CH_2_ and HCCO have more than one vibrational mode, we chose vibrational energy as the abscissa to distinguish each vibrational state. In the R2 channel, the vibrational state distribution of CO can be characterized by a Boltzmann plot (see Figure 7), compared with previous experiment results by Shaub et al. [31] and Chikan et al. [20]. Interestingly, the calculated results at *E*_c_ = 8.22, 13.05 kcal/mol and experimental results under different conditions have a similar distribution. Figure 8 shows the vibrational state distribution of CH_2_. The product vibrational state distribution presents the same feature from different initial states. The vibrational distribution of C-H bending (*ν*00), symmetric stretching (0*ν*0) and asymmetric stretching (00*ν*) modes represent the Boltzmann distribution. The distribution is “thermal-like”, and vibrational temperature depends on the available energy of different vibrational modes. The vibrational state distributions of the HCCO radical are presented in Figure 9. For the energy difference of HCCO bending modes is small, the vibrational distribution is very dense. The channels to higher-energy vibrational states are open, with low populations (less than 0.1%). When the C_2_H_2_ reactant is in the vibrational ground state, the vibrational state distributions of HCCO show a tendency of equilibrium, which is similar to the negative exponential distribution. When it comes to *ν*_1_ excited initial state, the distribution does not show a Boltzmann-like distribution.

## 4. Conclusions

To summarize, we report a detailed dynamic study for the O(^3^P) + C_2_H_2_ reaction on a recently developed, global, full-dimensional PIP-NN PES using the QCT approach. Due to the multi-channels and deep wells along the reaction pathways, the title reaction has complex dynamics. From the perspective of mode specificity, the translation of reactants can promote reactivity greatly. In contrast, vibrational excitation for all modes of C_2_H_2_ is ineffective. The calculated DCSs for both channels indicated a complex-forming mechanism, and the approaches of decomposition at the transition states led to the differences of DCSs between two channels. Due to the energy redistribution in the potential wells, the vibrational state distributions of all modes of products presented a Boltzmann-like distribution. In addition, all the results were compared with the experimental values in detail, which explains possible causes for the controversial results.

## Figures and Tables

**Figure 1 molecules-27-00754-f001:**
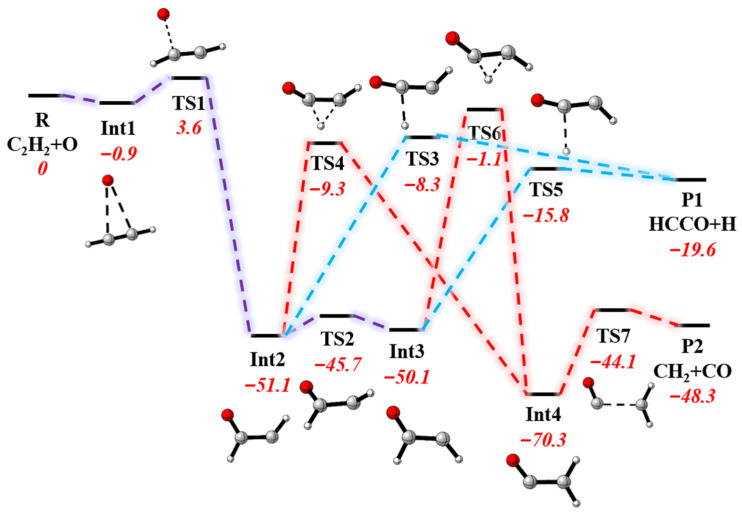
Schematic illustration of the pathways for the O + C_2_H_2_ reaction on the ground electronic state surface; energies are relative to the reactant asymptote in kcal/mol and within zero-point energy correction.

**Figure 2 molecules-27-00754-f002:**
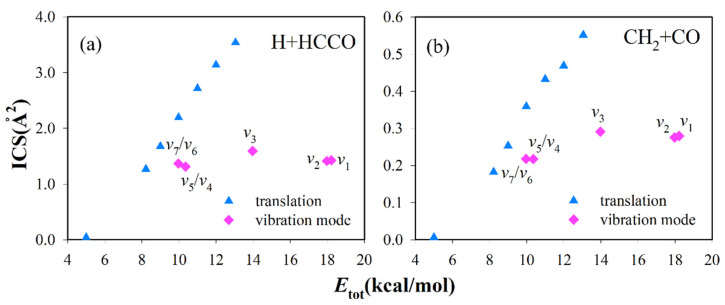
The integral cross-sections for (**a**) R1 and (**b**) R2 at collision energies ranging from 5 kcal/mol to 13.05 kcal/mol together with vibration modes excitation of C_2_H_2_ at *E*_c_ = 8.22 kcal/mol. *ν*_1_, *ν*_2_, *ν*_3_, *ν*_4_, *ν*_5_, *ν*_6_ and *ν*_7_ indicate as the first excitation for each mode of C_2_H_2_.

**Figure 3 molecules-27-00754-f003:**
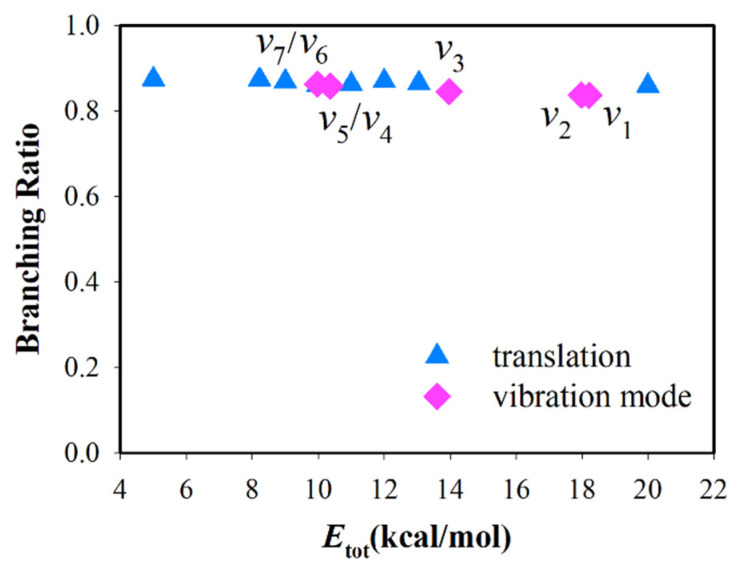
Branching ratio $\sigma_{R1}/\left(\sigma_{R1}+\sigma_{R2}\right)$ for the reaction O(^3^P) + C_2_H_2_ at collision energies ranging from 5 kcal/mol to 13.05 kcal/mol together with vibration modes excitation of C_2_H_2_ at *E*_c_ = 8.22 kcal/mol.

**Figure 4 molecules-27-00754-f004:**
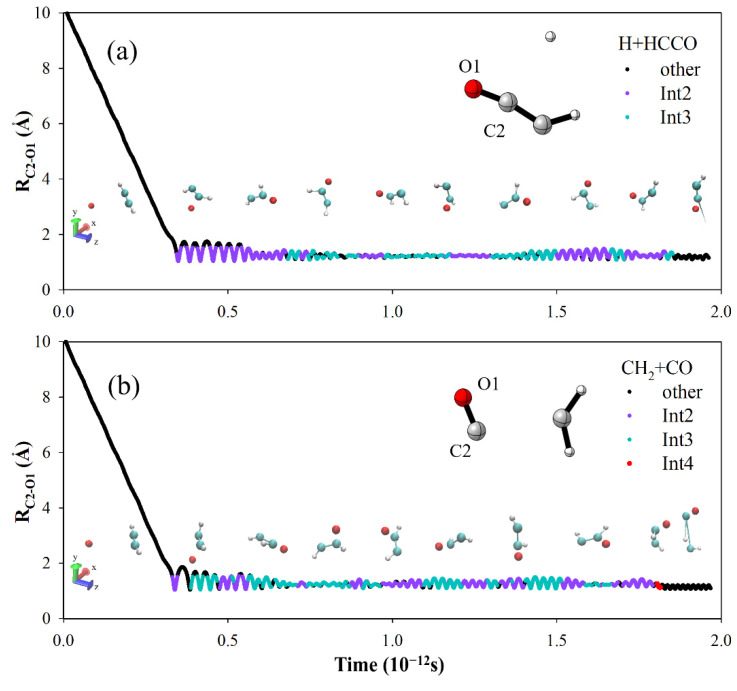
Bond distances as a function of reaction time in a trajectory for (**a**) the HCCO + H and (**b**) the CH_2_ + CO channel at *E*_c_ = 8.22 kcal/mol. We used geometry parameters (bond distances and angles) to distinguish Int2, Int3, and Int4 and show these complexes in different colors in trajectories.

**Figure 5 molecules-27-00754-f005:**
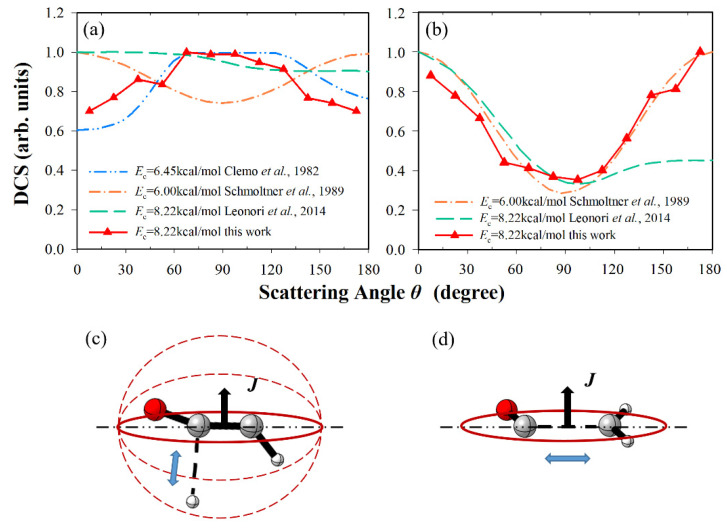
Differential cross-sections for (**a**) HCCO + H and (**b**) CH_2_ + CO channel, comparing with experiment results [10,14,22]. The maximum distribution is scaled to be 1. (**c**) The schematic diagram of TS5→H + HCCO. (**d**) The schematic diagram of TS7→CH_2_ + CO.

**Figure 6 molecules-27-00754-f006:**
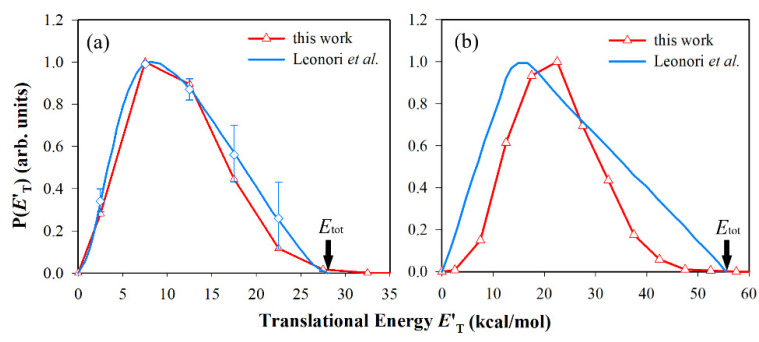
Product translational energy distributions for (**a**) HCCO + H and (**b**) CH_2_ + CO channels at *E*_c_ = 8.22kcal/mol, comparing with experiment results [22]. The maximum distribution is scaled to be 1.

**Figure 7 molecules-27-00754-f007:**
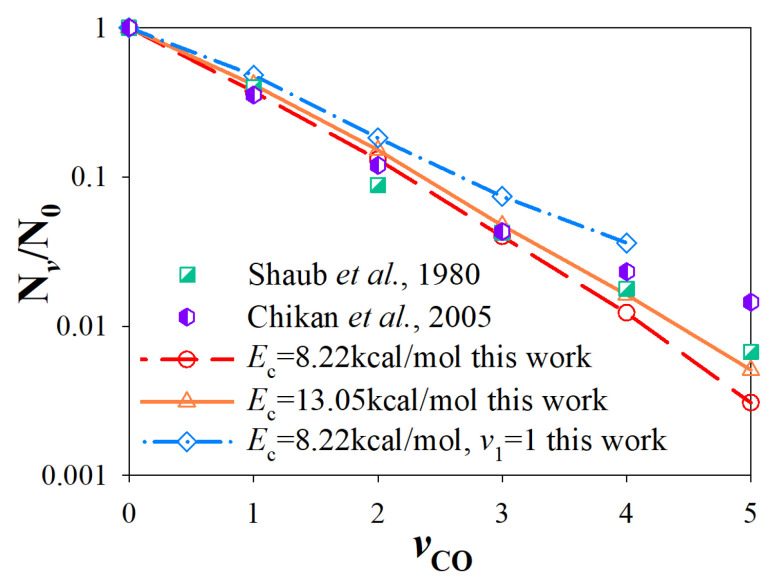
Vibrational state distributions of CO for the CO + CH_2_ channel from different initial states, compared with experiment results [20,31].

**Figure 8 molecules-27-00754-f008:**
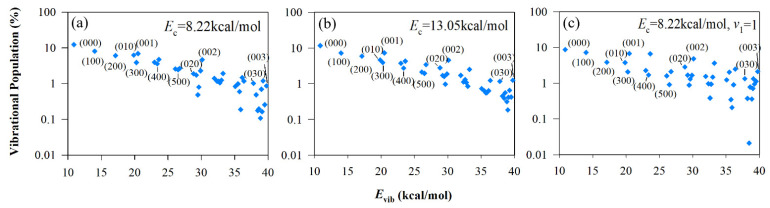
Vibrational state distributions of CH_2_ for the CO+CH_2_ channel from different initial states: (**a**) *E*_c_ = 8.22 kcal/mol; (**b**) *E*_c_ = 13.05 kcal/mol; (**c**) *E*_c_ = 8.22 kcal/mol and C-H symmetrical stretching mode (*ν*_1_) of C_2_H_2_ is excited at the first excitation.

**Figure 9 molecules-27-00754-f009:**
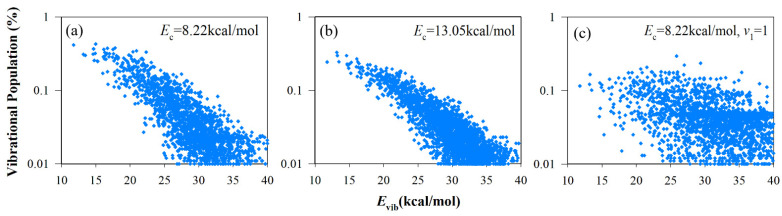
Vibrational state distributions of HCCO for the HCCO + H channel from different initial states: (**a**) *E*_c_ = 8.22 kcal/mol; (**b**) *E*_c_ = 13.05 kcal/mol; (**c**) *E*_c_ = 8.22 kcal/mol and C-H symmetrical stretching mode (*ν*_1_) of C_2_H_2_ is excited at the first excitation.

**Table 1 molecules-27-00754-t001:** SVP projections for the reactant modes of the O + C_2_H_2_ reaction at TS1.

Mode	Frequencies (cm^−1^)	SVP Values
Translation	-	0.723
*ν*_1_ (C-H symmetrical stretching)	3499.1	0.056
*ν*_2_ (C-H asymmetrical stretching)	3413.7	0.038
*ν*_3_ (C≡C stretching)	2011.3	0.045
*ν*_4_ and *ν*_5_ (symmetrical bending)	746.8	0.122
*ν*_6_ and *ν*_7_ (asymmetrical bending)	613.4	0.441

**Table 2 molecules-27-00754-t002:** Product energy fraction for the reaction O(^3^P) + C_2_H_2_→HCCO + H at *E*_c_ = 6.00, 8.22, 13.05 kcal/mol and mode excitation of C_2_H_2_ at *E*_c_ = 8.22 kcal/mol.

*E* _c_	Vibrational Excitation	Total Energy	*f_tran_* ^a^	*f_rot_* ^a^	*f_rot_* (HCCO) ^a^	*f_vib_* (HCCO) ^a^
6.00	-	6.00	41.53% (41.7% ^b^)	0.43%	13.39%	44.65%
8.22	-	8.22	39.54% (42% ^c^)	0.47%	13.98%	46.01%
13.05	-	13.05	36.75% (35% ^c^)	0.54%	17.39%	45.33%
8.22	*ν*_1_ = 1	18.22	34.88%	0.44%	11.57%	53.11%
8.22	*ν*_2_ = 1	17.98	34.55%	0.44%	11.74%	53.27%
8.22	*ν*_3_ = 1	13.97	36.91%	0.46%	13.02%	49.61%
8.22	*ν*_4_ = 1/*ν*_5_ = 1	10.36	38.65%	0.48%	13.30%	47.57%
8.22	*ν*_6_ = 1/*ν*_7_ = 1	9.97	38.95%	0.46%	13.81%	46.78%

^a^*f_tran_*, *f_rot_* and *f_vib_* mean fraction of energy as relative translation, rotation, and vibration, respectively. ^b^ The results from crossed molecular beam experiments of Schmoltner et al. [14]. ^c^ The results from crossed molecular beam experiments of Leonori et al. [22].

**Table 3 molecules-27-00754-t003:** Product energy fraction (%) for the reaction O(^3^P)+C_2_H_2_→CH_2_+CO at *E*_c_ = 6.00, 8.22, 13.05 kcal/mol and mode excitation of C_2_H_2_ at *E*_c_ = 8.22 kcal/mol.

*E* _c_	Vibrational Excitation	Total Energy	*f_tran_* ^a^	*f_rot_* ^a^	*f_rot_* (CO) ^a^	*f_vib_* (CO) ^a^	*f_rot_* (CH_2_) ^a^	*f_vib_* (CH_2_) ^a^
6.00	-	6.00	40.36% (41.3% ^b^)	0.47%	13.68%	6.47%	19.86%	19.17%
8.22	-	8.22	39.40% (42% ^c^)	0.48%	14.26%	4.92%	16.99%	23.95%
13.05	-	13.05	37.96% (42% ^c^)	0.59%	14.36%	5.04%	17.34%	24.72%
8.22	*ν*_1_ = 1	18.22	35.88%	0.45%	13.52%	5.83%	16.96%	27.35%
8.22	*ν*_2_ = 1	17.98	36.30%	0.48%	14.67%	6.05%	17.24%	25.27%
8.22	*ν*_3_ = 1	13.97	37.45%	0.51%	14.06%	5.88%	17.27%	24.82%
8.22	*ν*_4_ = 1/*ν*_5_ = 1	10.36	39.03%	0.49%	14.29%	4.65%	16.89%	24.65%
8.22	*ν*_6_ = 1/*ν*_7_ = 1	9.97	38.10%	0.44%	14.41%	5.67%	17.24%	24.15%
-	-	-	-	-	-	(7.4% ^d^)	-	-
-	-	-	-	-	-	6.1 ± 1.3% ^e^)	-	-

^a^*f_tran_*, *f_rot_* and *f_vib_* mean, respectively, fraction of energy as relative translation, rotation, and vibration. ^b^ The results from crossed molecular beam experiments of Schmoltner et al. [14]. ^c^ The results from crossed molecular beam experiments of Leonori et al. [22]. ^d^ The results from FTIR emission experiments of Chikan and Leone [20]. ^e^ The results from CO laser absorption experiments of Shaub et al. [31].

## Data Availability

No applicable.

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
