# Peer review of "Dissection of the Multichannel Reaction O(3P) + C2H2: Differential Cross-Sections and Product Energy Distributions"

_molecules, 2022, doi:10.3390/molecules27030754_

Round 1
Reviewer 1 Report
attached

Reviewer 2 Report
This paper complements (perhaps completes) an earlier QCT study on O + HCCH reaction. This reaction is very interesting and important and has been approached by many in the molecular beam community spanning nearly 4 decades now, with still some questions remaining as to mechanism. This study aims (and convincingly succeeds I believe) to answer some of the outstanding questions for this system. I find it to be well done, and a fine addition to the literature on this system.
I recommend publication nearly as is, once a good check of grammar is done. I list just a few slight suggestions for the authors to think consider.
ln 56 really should also cite 19 and 21 here as CMB exps. even though you will not talk about them more.
ln 63 "Anyway,... " seems a strange transition.
ln 195-198 I found myself wanting more information/discussion here, perhaps with some specifics referencing the PES. Consider expanding a bit.
Fig. 4b put legend in same order as 4a
awkward/check grammar ln 13, 61, 79, 208, 236, 239, 247, 250
Why no data points and error bars for Leonori in Fig 2b?
It is unfortunate that the v6/v7 excitation (most coupled per SVP) was not also tried at the lower translational energy- perhaps the lower translation energies are better to see vibrational enhancement effect, if any.
Section 3.3 seems short and rushed. There is much data here, is there nothing else the authors want to say about the vibrational distributions?
Reviewer 3 Report
This manuscript presents a quasi-classical trajectory study on the O(³P)+C2H2 reaction, using the VENUS program and a neural-network potential energy surface previously developed by some of the authors (in ref 29). The branching ratio between the CH2+CO / H +HCCO products is analyzed for several translational energies and different excitations of the reactants. The small effect of the initial vibrational excitation is analysed using the sudden vector projection technique, and it is concluded that this is so bacause long-living complexes are formed which erase the information of the reactants. Differential cross sections are also calculated for the two product chgannels, in good agreement with experimental results. Also the final product energy distribution is presented.
The methods used are up-to-date and are in general well described. The results are interesting, but is not clear what is new with respect to the work previously presented in Ref. 29. I consider this work as publishable in Molecules, but some modifications must be done prior to acceptance, as described below:
1) The differences with the previous results presented in Ref.29 and some others must be outlined, indicating in the conclusions the novelty of this work.
2) More physical insight need to be given in the manuscript, which is rather descriptive, and based on the analysis of the final results, without going in details of the dynamics and reaction mechanism. For example, it is argued that the influence of the initial vibrational excitations of reactants is erased by the formation of long-lived complexes, withour providing any further results. This argument should be supported by the analysis of the lifetime of the complex, and where it expendes longer time, int2, int3 or already in Int4
3) Long lived complexes are also used to argue the large vibrational excitation of products. Furthermore, the nearly isotropic DCS for HCCO products and the forward/backward DCS for CH2+CO, is attributed to some schematic arguments based (unclear to my understanding) based on the limiting TS5 and TS7 structures. These models need to be further explained and supported by the analysis of the dynamics.
4) The reaction mechanism is therefore unclear and need to be clarified by the analysis of the trajectories. Do the authors mean that a long-lived complex is formed in the int2/int3 region (loosing the information of the initial state), followed by a bifuraction to either of the two products through Ts5 and TS7?. This could be clarified by showing some characteristic trajectories.
5) Also, why the authors use GB for polyatomic fragments and HB for diatomic products. Is simply to increase the statistics, otherwise very poor?
6) In the first paragraph of section 3 (before subsection 3.1) tha authors says "... a lot od available energy will flow into other internal degrees of freedom of the complex via rapid trans-cis isomerization." The isomerization is between int2 and int3?
7) In the discussion of the products vibrational excitation, in Figs. 7 and 8, the modes of CH2 and HCCO need to be defined. Moreover, in this last case, is there any particular pattern for each vibrational mode of HCCO?
